# Long term temporal trends in synoptic-scale weather conditions favoring significant tornado occurrence over the central United States

**Mohamed Elkhouly[1], Stephanie E. Zick[2]\*, Marco A. R. Ferreira[3]**

**1** Department of Mathematics and Statistics, Cleveland State University, Cleveland, OH, United States of America, **2** Department of Geography, Virginia Tech, Blacksburg, VA, United States of America, **3** Department of Statistics, Virginia Tech, Blacksburg, VA, United States of America

\* sezick@vt.edu

**Data Availability Statement:** Software and data supporting this study are available at https://doi.org/10.7294/21750689.

## Abstract

We perform a statistical climatological study of the synoptic- to meso-scale weather conditions favoring significant tornado occurrence to empirically investigate the existence of long term temporal trends. To identify environments that favor tornadoes, we apply an empirical orthogonal function (EOF) analysis to temperature, relative humidity, and winds from the Modern-Era Retrospective analysis for Research and Applications Version 2 (MERRA-2) dataset. We consider MERRA-2 data and tornado data from 1980 to 2017 over four adjacent study regions that span the Central, Midwestern, and Southeastern United States. To identify which EOFs are related to significant tornado occurrence, we fit two separate groups of logistic regression models. The first group (LEOF models) estimates the probability of occurrence of a significant tornado day (EF2-EF5) within each region. The second group (IEOF models) classifies the intensity of tornadic days either as strong (EF3-EF5) or weak (EF1-EF2). When compared to approaches using proxies such as convective available potential energy, our EOF approach is advantageous for two main reasons: first, the EOF approach allows for the discovery of important synoptic- to mesoscale variables previously not considered in the tornado science literature; second, proxy-based analyses may not capture important aspects of three-dimensional atmospheric conditions represented by the EOFs. Indeed, one of our main novel findings is the importance of a stratospheric forcing mode on occurrence of significant tornadoes. Other important novel findings are the existence of long-term temporal trends in the stratospheric forcing mode, in a dry line mode, and in an ageostrophic circulation mode related to the jet stream configuration. A relative risk analysis also indicates that changes in stratospheric forcings are partially or completely offsetting increased tornado risk associated with the dry line mode, except in the eastern Midwest region where tornado risk is increasing.

**Funding:** This study was financially supported by the Virginia Tech Libraries Open Access Subvention Fund in the form of an award to cover article processing fees. No additional external funding was received for this study. The funder had no role in study design, data collection and analysis, decision to publish, or preparation of the manuscript.

**Competing interests:** The authors have declared that no competing interests exist.

## Introduction

Tornadoes associated with severe thunderstorms cause damage to property and loss of life [1]. Although tornado frequency shows obvious interannual variability, the average number of strong tornadoes (EF3-EF5) in the United States (U.S.) has remained mostly stable [2, 3]. Spatially, negative trends in tornado frequency have been observed in the central and southern Great Plains region of the U.S. and positive trends in the Midwestern and Southeastern U.S. [4–8]. Other researchers have found that increased tornado frequency in the Southeastern U.S. is partially explained by a coinciding increase in population density, while variability in the U.S. Plains region are partially explained by the advancements in radar technology [3]. Even though there is some uncertainty about regional trends in tornado numbers, there is broader consensus that the frequency of U.S. outbreaks with multiple tornadoes is increasing, especially for the most extreme tornado outbreaks [2, 9, 10]. Moore et al. (2017) [11] also found an increase in high-frequency tornado days, mostly due to an increasing number of outbreaks in regions east of the Great Plains.

Evidence of the impact of climate change on tornado occurrence remains unclear [10, 12–14]. While reanalysis datasets show consistent warming trends throughout the troposphere [15], there is more uncertainty in how this warming has and will continue to impact the atmospheric conditions that favor severe convective environments. Severe thunderstorms are known to occur in environments with specific kinematic and thermodynamic characteristics including high convective available potential energy (CAPE), high deep layer shear (DLS) and enhanced storm relative helicity (SRH) [16, 17]. To study the impact of climate change on severe thunderstorms, some researchers have used projections from numerical climate models that consider possible future scenarios of environmental proxies such as CAPE, DLS, SRH, and convective precipitation (CP) [18–24]. For example, Coupled Model Intercomparison Project Phase 6 (CMIP6) data suggest increases in favorable convective environments broadly but with considerable regional variability [24]. However, long-term trends in the historical frequency of convective proxies based on reanalysis data have garnered mixed results and a high degree of regional variability [24–26]. Importantly, that regional variability is generally consistent across studies, with numerous findings that the large-scale environment in the southeastern U.S. is becoming more favorable for severe weather [25, 26].

Studying the future impact of climate change using environmental proxies based on reanalysis data is under scrutiny for two main reasons. First, these proxies vary spatially [19, 20, 27, 28]. While reanalyses and climate models can generally simulate the wind field, there are known regional biases in their representation of the thermodynamic environment [29]. Taszarek et al. (2021) reveal disagreements between ERA5 and rawinsonde observations over some regions, and suggest to use caution when interpreting results for any one region based on a single dataset [25]. To date, most retrospective studies have used the North American Regional Reanalysis (NARR) [10, 26, 30] or, more recently, the ECMWF Reanalysis version 5 (ERA5) [31, 32] to assess trends in severe weather environments using proxies. Due to regional biases and overall uncertainties in reliability of a single dataset, it is important to look at historical trends in severe weather environments in multiple high quality reanalyses. Second, models that depend on proxies usually assume unchanged association between occurrence of severe thunderstorms and proxies [19, 20, 27, 33] although that empirical association may change due to climate change [34]. For these reasons, long term trends in the frequency of environments supportive of severe thunderstorms are still unclear.

Instead of using proxies, here we perform a statistical climatological study of synoptic-scale weather conditions favoring significant tornado occurrence over the Central,

Midwestern, and and Southeastern United States to empirically investigate the existence of long term temporal trends. To identify synoptic-scale weather conditions, we take an empirical orthogonal function (EOF) approach applied to the meteorological variables of air temperature, relative humidity, and wind components (zonal wind, meridional wind, and vertical velocity (omega, Pa/s)) from the Modern-Era Retrospective analysis for Research and Applications Version 2 (MERRA-2) dataset (see Materials and methods section). Specifically, we consider MERRA-2 data and tornado data from 1980 to 2017 over four adjacent study regions with high rates of tornado occurrence [35–38]. Compared with other reanalyses such as the NARR, MERRA-2 focuses more on the stratosphere, especially the representation of stratospheric ozone. Due to the relationships between the upper troposphere and favorable tornado environments [39–41], it is important for the reanalysis to capture stratosphere-troposphere interactions. No previous studies have analyzed trends in severe convective environments based on MERRA-2. Because of overall uncertainties in reanalysis representation of convective environments and their trends, it is important to consult other high-quality reanalyses, such as MERRA-2, so that research is not biased by the results from any one reanalysis. While MERRA-2 is known to have reduced CAPE compared with observations [42], this is a common feature across many reanalysis datasets [29, 31, 42]. Overall, differences in MERRA-2 temperature and winds compared with a reanalysis mean are negligible below 10 mb [43], and MERRA-2 successfully discriminates between significant and weak tornado environments [42]. More information on MERRA-2 is available in the second section.

Our data-driven EOF approach identifies three dimensional atmospheric signals that manifest in the temperature, relative humidity and wind fields. These signals can be interpreted as synoptic-scale and meso-alpha scale "ingredients" associated with tornado occurrence. Ingredients-based approaches [44] are frequently used to identify favorable convective environments (e.g., drylines [45–47], low-level jets [48–50], and warm sectors of extratropical cyclones [51, 52]. Other studies have used ingredient-based approaches to study tornadoes and severe thunderstorms [53–57]. Numerous previous studies have used EOF analysis to identify synoptic-scale patterns that favor tornado development [58–62].

To identify which EOFs are related to significant tornado occurrence, we fit two separate groups of logistic regression models. The first group of models (LEOF models) estimates the probability of occurrence of a significant tornado day (EF2-EF5) within each region. The second group of models (IEOF models) classifies the intensity of tornadic days either as strong (EF3-EF5) or weak (EF1-EF2). From the fits of the LEOF and IEOF models to each of the four study regions, we identify EOF modes that are significantly related to tornado occurrence. When compared to approaches based on proxies such as CAPE, our EOF approach is advantageous for two main reasons: first, the EOF approach allows for the discovery of important synoptic-scale signals that may not have been considered previously in the tornado science literature; second, proxy-based analyses may not capture important aspects of three-dimensional configurations represented by the EOFs.

For the identified significant EOF modes, we investigate the existence of long-term temporal trends. Specifically, we find long-term temporal trends in three EOF modes: the stratospheric forcing mode (STF), a dry line mode (DLM), and an ageostrophic circulation wind mode (ACM) related to the jet stream configuration. In addition, to investigate the impact of temporal changes in these EOF modes on the probability of a significant tornado day, we perform a study of conditional relative risk (RR) [63]. This conditional relative risk study shows that, when compared to changes in the wind ACM mode, the long term trends observed in the DLM and STF modes have had a much stronger effect on the risk of significant tornado days.

## Materials and methods

### Data

To evaluate the atmospheric environment, we utilize data from four adjacent regions over the Central, Midwestern and Southeastern United States. These four regions have the highest rates of tornado occurrence per square mile [35–37]. Fig 1 shows the four study regions (R1, R2, R3 and R4) bounded by 100–84.375˚W and 32–43.5˚N. Moore et al. (2022) used a similar four-domain region [26]. These domains include the southern Plains (R1), Southeast (R2), western Midwest (R3) and eastern Midwest (R4) regions of the U.S. Each of the four regions has 156 grid cells (13 longitudes × 12 latitudes on a 0.625˚ × 0.5˚ grid) from the Modern-Era Retrospective analysis for Research and Applications, Version 2 (MERRA-2) dataset [64] with 3-hour temporal resolution for the years 1980–2017. The MERRA-2 dataset is available through NASA Goddard Earth Sciences Data Information Services Center (https://doi.org/10.5067/VJAFPLI1CSIV). We consider the variables of air temperature, relative humidity, zonal wind, meridional wind, and vertical velocity (omega, Pa/s). Specifically, we utilize meteorological output from 28 vertical pressure levels from 925 hPa to 10 hPa.

MERRA-2 is a high-quality, next-generation reanalysis that includes the stratosphere and assimilates aerosol data. Previous tornado climatology studies have used the NARR [10, 26, 30], which is a higher resolution dataset but only includes the lower stratosphere. Compared with other reanalyses, there is more emphasis on stratospheric ozone and aerosol information. In the ERA5, the evolution of tropospheric aerosols is based on data from CMIP5, while MERRA-2 assimilates actual aerosol data [43]. Overall, differences in MERRA-2 temperature and winds compared with a reanalysis mean are negligible below 10 mb [43]. Stratospheric-tropospheric interactions are generally handled authentically [65, 66] (e.g., that the tropospheric inversion layer is strengthened following a sudden stratospheric warming event [67]. MERRA-2 and ERA5 temperature and wind climatologies are similar in the stratosphere, and trends are similar in the lower stratosphere (below 10 mb) [68]. Other studies have examined the MERRA-2 representation of convective environments and found that ERA5

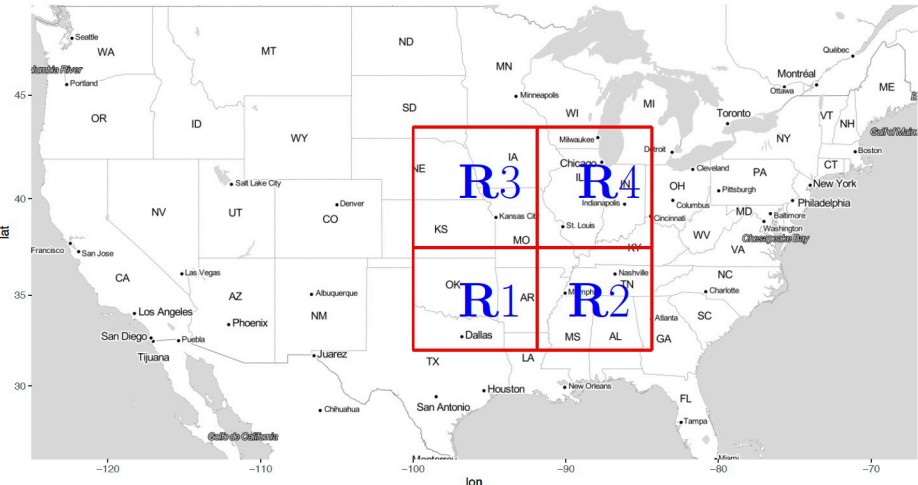

**Fig 1. Map of the United States with the red box indicating the study region (-100:-84.375 longitude and 32:43.5 latitude on 0.625˚×0.5˚ grid).** The first region (R1) is bounded by -100:-92.5 longitude and 32:37.5 latitude. The second region (R2) is bounded by -91.875:-84.375 longitude and 32:37.5 latitude. The third region (R3) is bounded by -100:-92.5 longitude and 38:43.5 latitude. Lastly, the fourth region (R4) is bounded by -91.875:-84.375 longitude and 38:43.5. Each of the four regions contains 156 spatial positions of the weather MERRA-2 data (13 longitudes × 12 latitudes).

**Table 1. Number of significant, weak and strong tornado days for each of the four regions from 1980 to 2017.**

| Region | Significant tornado days (EF2-EF5) | Incidence rate of significant tornado days (%) | Weak tornado days (EF1-EF2) | Strong tornado days (EF3-EF5) |
|---|---|---|---|---|
| R1 | 335 | 2.42 | 632 | 135 |
| R2 | 333 | 2.40 | 567 | 132 |
| R3 | 253 | 1.82 | 513 | 101 |
| R4 | 254 | 1.83 | 572 | 90 |

generally produces slightly higher correlations with rawindsonde observations, [32] but both datasets provide similar trends in both the dynamic and thermodynamic fields [15]. Both ERA5 and MERRA-2 struggle with near surface parameters such as low-level lapse rates and wind shear [32]. However, importantly, MERRA-2 successfully discriminates between significant tornado and weak tornado environments [42]. Overall, MERRA-2 is a high quality reanalysis that is an appropriate choice for deriving information about the synoptic scale environment. One important caveat is that stratospheric moisture should be considered with caution [43].

In this study, we use tornado data provided by the National Oceanic and Atmospheric Administration (NOAA) and Storm Prediction Center (SPC) (available online at http://www. spc.noaa.gov/wcm/#gis). We focus on two groups of models. The first group estimates the probability of occurrence of a significant tornado day (EF2-EF5) within each of the four study regions. The second group of models predicts the intensity of tornadic days as either weak (EF1-EF2) or strong (EF3-EF5). Table 1 summarizes the total number of significant, weak and strong tornado days for each of the four regions from 1980 to 2017.

To validate our proposed models for estimating tornado probability and predicting tornadic intensity, we consider competing models with other more traditional explanatory variables for tornado occurrence. Those variables are convective available potential energy (CAPE), convective precipitation (CP), storm relative helicity (SRH), convective inhibition (CIN), and best (4-layer) lifted index (4LFTX) provided by the North American Regional Reanalysis (NARR) (available at https://psl.noaa.gov/data/gridded/data.narr.html). In addition, we consider Global Wind Oscillation (GWO) phases provided by NOAA Physical Sciences Laboratory (available at https://psl.noaa.gov/map/clim/gwo.data.txt) and the monthly detrended Niño-3.4 index provided by the NOAA Climate Prediction Center (available at https://www. cpc.ncep.noaa.gov/products/analysis_monitoring/ensostuff/detrend.nino34.ascii.txt).

## Empirical orthogonal function (EOF) analysis

To reduce the massive number of predictors from the MERRA-2 dataset, we identify atmospheric EOF modes for the study regions. To obtain interpretable EOF modes, we consider two groups of meteorological variables: the first group contains air temperature and relative humidity (TRH), and the second group contains the three wind components (WC). For each of the four regions, we consider the correlation matrix of each group of meteorological variables. Each correlation matrix measures the dependence amongst the considered variables within that corresponding region. We find that the correlation matrices of each group of meteorological variables are similar across the four regions (R1, R2, R3 and R4). Thus, to construct EOF modes with unified interpretations over the four regions, we obtain common loadings for the four regions as follows. First, we obtain the maximum likelihood estimator (MLE) of the correlation matrix of each group of variables by averaging the corresponding correlation matrices calculated for each of the four regions. This results in only two correlation matrices: one correlation matrix for air temperature and relative humidity and another

**Table 2. Summary of the explained variance, proportion of variability explained by each temperature and relative humidity mode as well as the cumulative proportions.** The $Temp - Rhum_i$ is the $i^{th}$ EOF mode of the air temperature and relative humidity group.

| EOF mode | Variance explained | Proportion explained | Cumulative proportion |
|---|---|---|---|
| $Temp - Rhum_1$ | 3318.91 | 0.38 | 0.38 |
| $Temp - Rhum_2$ (STF) | 1031.05 | 0.12 | 0.50 |
| $Temp - Rhum_3$ | 886.85 | 0.10 | 0.60 |
| $Temp - Rhum_4$ | 477.86 | 0.05 | 0.65 |
| $Temp - Rhum_5$ (MSS) | 233.17 | 0.03 | 0.68 |
| $Temp - Rhum_6$ (DSS) | 209.96 | 0.02 | 0.70 |
| $Temp - Rhum_7$ (DLM) | 205.35 | 0.02 | 0.73 |
| $Temp - Rhum_8$ | 172.66 | 0.02 | 0.75 |
| $Temp - Rhum_9$ | 144.96 | 0.02 | 0.76 |
| $Temp - Rhum_{10}$ | 124.10 | 0.01 | 0.78 |
| $Temp - Rhum_{11}$ | 108.16 | 0.01 | 0.79 |
| $Temp - Rhum_{12}$ | 95.26 | 0.01 | 0.80 |
| $Temp - Rhum_{13}$ | 85.38 | 0.01 | 0.81 |
| $Temp - Rhum_{14}$ | 66.26 | 0.01 | 0.82 |
| $Temp - Rhum_{15}$ | 61.15 | 0.01 | 0.83 |

correlation matrix for the three wind components. Next, we perform the EOF analyses [69] separately for each of the two correlation matrices. This leads us to retain 15 atmospheric EOF modes from each group that explain respectively about 83% and 69% of the total variability of the TRH and WC variables (Tables 2 and 3). Thus, we have EOF modes with common interpretation (the same loadings) over the four regions capturing the dependence among the original variables. In addition, each of the four regions has its own calculated EOF modes. That is, the loadings are the same across the study regions but the calculated EOFs are specific to each region. Those EOF modes are data-driven, non subjective indices where each EOF mode is a three-dimensional weighted average of the meteorological

**Table 3. Summary of the explained variance, proportion of variability explained be each wind component mode as well as the cumulative proportions.** The $Wind_i$ is the $i^{th}$ EOF mode of the wind group.

| EOF mode | Variance explained | Proportion explained | Cumulative proportion |
|---|---|---|---|
| $Wind_1$ (HSD) | 2589.79 | 0.20 | 0.20 |
| $Wind_2$ | 2300.16 | 0.18 | 0.37 |
| $Wind_3$ | 839.84 | 0.06 | 0.44 |
| $Wind_4$ | 688.54 | 0.05 | 0.49 |
| $Wind_5$ | 579.85 | 0.04 | 0.53 |
| $Wind_6$ | 477.42 | 0.04 | 0.57 |
| $Wind_7$ | 356.08 | 0.03 | 0.60 |
| $Wind_8$ | 233.17 | 0.02 | 0.62 |
| $Wind_9$ | 184.42 | 0.01 | 0.63 |
| $Wind_{10}$ (ACM) | 150.80 | 0.01 | 0.64 |
| $Wind_{11}$ | 142.56 | 0.01 | 0.65 |
| $Wind_{12}$ | 128.37 | 0.01 | 0.66 |
| $Wind_{13}$ | 125.66 | 0.01 | 0.67 |
| $Wind_{14}$ | 114.06 | 0.01 | 0.68 |
| $Wind_{15}$ | 110.25 | 0.01 | 0.69 |

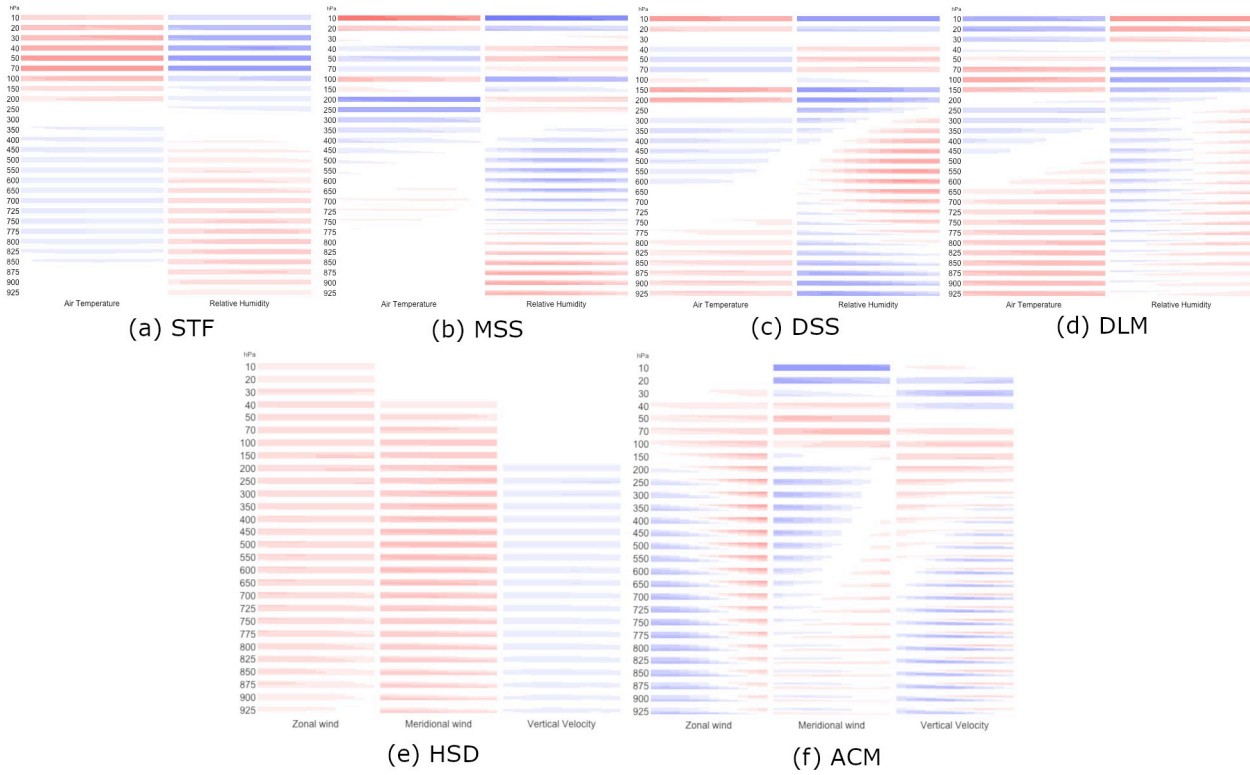

**Fig 2. Plots for the spatial loadings of important weather EOFs at different altitude pressure levels.** Within each panel, each column corresponds to one weather variable and each row corresponds to one altitude pressure level. Each rectangle maps spatially the 182 loadings, one for each MERRA-2 data location in Fig 1. Colors indicate sign and magnitude of the loadings: red for positive and blue for negative loadings; darker colors for larger magnitude, and white for negligible loadings. (a) STF, (b) MSS, (c) DSS, (d) DLM, (e) HSD, and (f) ACM.

variables (TRH or WC). The corresponding weights are the loadings determined according to the dependence of the considered variables on the three dimensions (longitude, latitude and altitude) as shown in Fig 2. In what follows, within each region, we study the possible impact of the 30 atmospheric modes on occurrence of a significant tornado day (a day with an EF2-EF5 tornado) in addition to another group of models predicting the intensity of tornadic days as either weak or strong.

## The logistic EOF (LEOF) regression model for occurrence of a significant tornado day

To increase the statistical capability of detecting relationships between meteorological variables and occurrence of a significant tornado, we downsample the non-tornado days [70, 71]. For each region, we use a case-control framework to raise the incidence rate of significant tornado days to about 10% by selecting, when possible, nine control days for each corresponding tornado (case) day. Specifically, a control day for a corresponding tornado day is the same day of the same month but in a different year. In addition, a control day does not have any recorded tornado activity of any intensity in that region.

To increase the likelihood that the predictors are associated with an atmospheric environment that precedes a tornado rather than an environment resulting from occurrence of the tornado (or the parent storm), we build our statistical models with EOF modes at the time step

before the first registered significant tornado on each significant tornado day in our study regions. For example, if the first registered significant tornado was at 1300 UTC in a specific region, we use the available EOF modes and other meteorological variables at 1200 UTC for the tornado day and the corresponding control days.

In addition to the 30 atmospheric EOF modes, we also consider a few additional predictors. First, to account for tropopause pressure level, we run EOF analysis for the thermal tropopause from the MERRA-2 dataset. That tropopause EOF analysis is similar to the EOF analysis conducted for the meteorological variables. We retain two tropopause EOF modes explaining 80% of the total variability of tropopause data over our study region. Similarly, the two tropopause EOF modes have the same interpretation across study regions but the calculated EOF modes are specific to each of the four study regions.

Next, to account for the influence of the El Niño Southern Oscillation (ENSO) that affects tornado occurrence in the United States [72], we include the monthly detrended Niño-3.4 index. Further, to estimate seasonal effects we use one indicator variable (seasonal dummy) for each month from January to November and keep December as the baseline month. Because of the possible nonlinearity in the relationship between tornado occurrence and the EOF predictors, in addition to the linear terms, we also consider quadratic terms of all predictors.

For each of the four study regions, let

$$Y_t = \begin{cases} 1, & \text{if } t \text{ is an EF2} - \text{EF5 tornado day in that region,} \\ 0, & \text{if } t \text{ is a control day.} \end{cases}$$

Then, the logistic regression model we consider is

$$logit(\pi_t) = \beta_0 + \sum_{k=1}^{P} \beta_k x_{tk}, \tag{1}$$

where $\pi_t = Pr(Y_t = 1 | X_t = x_t)$, $logit(\pi_t) = log(\pi_t/(1 - \pi_t))$, $\beta_k$ is the regression coefficient for predictor $k$, $X_t$ is the vector of predictors on day $t$, and $x_{tk}$ is the value of predictor $k$ on day $t$. Thus, the probability of a day with a touchdown of any EF2-EF5 tornado in that specific region equals to $\pi_t$. Exhaustive search for the best logistic model for each of the four regions is not computationally feasible because there are $2^{77} \approx 10^{23}$ possible models (77 predictors). To search for the best logistic model, we use a genetic algorithm (GA) implemented in the R package GA [73]. To select the best model we use the Bayesian Information Criterion (BIC), which allows the computation of approximate posterior probabilities for each model that are proportional to $e^{-0.5BIC}$ [74, 75]. We use the GA with 500 different initial models and we retain the same logistic model for each region (Table 4). Hereafter, we call these models the logistic EOF (LEOF) models.

## Models classifying intensity of tornadic days

In addition to the LEOF models, we introduce another group of models to classify the intensity of tornadic days. Specifically, for each region, we consider a logistic regression model to classify tornadic days as either weak (EF1-EF2) or strong (EF3-EF5). We utilize the same predictors used for the LEOF models, and similarly, we consider the data for the time step before the first tornado of the corresponding intensity. For each of the four regions, let

$$Z_t = \begin{cases} 1 & \text{if } t \text{ is an EF3} - \text{EF5 tornado day in that region,} \\ 0 & \text{if } t \text{ is an EF1} - \text{EF2 tornado day in that region.} \end{cases}$$

**Table 4. Summary of the fitted logistic regression models estimating probability of occurrence of a significant tornado (EF2-EF5) in each of the four regions.** For each region, the table has only the estimates of the significant predictors and their corresponding standard error (**Std. Error**).

| Coefficient (Std. Error) | R1 | R2 | R3 | R4 |
|---|---|---|---|---|
| Intercept | 6.921 (2.950) | 6.579 (2.642) | -6.325 (0.395) | -5.290 (0.305) |
| $Temp - Rhum_1$ | 0.034 (.004) | 0.048 (.004) | 0.052 (.005) | 0.053 (.004) |
| $Temp - Rhum_2$ (STF) | 0.044 (.005) | 0.023 (.004) | 0.022 (.005) | |
| $Temp - Rhum_2^2$ $(STF)^2$ | -0.0004 (.0001) | | | |
| $Temp - Rhum_3$ | 0.014 (.004) | 0.017 (.004) | | |
| $Temp - Rhum_5$ (MSS) | 0.030 (.007) | 0.020 (.006) | 0.049 (.010) | |
| $Temp - Rhum_6$ (DSS) | 0.031 (.006) | 0.032 (.006) | 0.033 (.009) | 0.028 (.008) |
| $Temp - Rhum_7$ (DLM) | 0.106 (.01) | 0.091 (.009) | 0.064 (.011) | 0.065 (.01) |
| $Temp - Rhum_8$ | 0.031 (.007) | | | |
| $Temp - Rhum_9$ | | | 0.051 (.010) | 0.043 (.010) |
| $Temp - Rhum_{11}$ | -0.034 (.010) | | | |
| $Temp - Rhum_{11}^2$ | | 0.0010 (.0003) | | |
| $Temp - Rhum_{12}^2$ | -0.003 (.001) | | | |
| $Temp - Rhum_{14}^2$ | -0.003 (.001) | | -0.003 (.001) | -0.005 (.002) |
| $Temp - Rhum_{15}^2$ | 0.003 (.001) | | | 0.045 (.013) |
| $Wind_1$ (HSD) | 0.039 (.002) | 0.048 (.005) | 0.044 (.004) | 0.041 (.002) |
| $Wind_1^2$ $(HSD)^2$ | | -0.0001 (<0.0001) | | |
| $Wind_3$ | | 0.018 (.004) | | |
| $Wind_4$ | -0.018 (.004) | | | |
| $Wind_5$ | -0.019 (.004) | | -0.024 (.005) | -0.021 (.005) |
| $Wind_6$ | | 0.021 (.005) | 0.025 (.005) | |
| $Wind_6^2$ | | 0.0005 (.0001) | | |
| $Wind_8$ | | | | 0.023 (.006) |
| $Wind_{10}$ (ACM) | | 0.026 (.006) | 0.024 (.008) | 0.027 (.008) |
| $Wind_{10}^2$ $(ACM)^2$ | | | 0.001 (.0003) | |
| $Wind_{11}$ | | 0.025 (.007) | | |
| Month June | | | -0.904 (.235) | |
| $Tropopause_1$ | -0.051 (.014) | | | |
| $Nino - 3.4$ | -0.433 (.110) | -0.439 (.098) | | |

Then, the logistic regression model we consider is

$$logit(\lambda_t) = \alpha_0 + \sum_{k=1}^{P} \alpha_k x_{tk}, \tag{2}$$

where $\lambda_t = Pr(Z_t = 1|X_t = x_t)$, $logit(\lambda_t) = log(\lambda_t/(1 - \lambda_t))$, $\alpha_k$ is the regression coefficient for predictor $k$, $X_t$ is the vector of predictors on day $t$, and $x_{tk}$ is the value of predictor $k$ on day $t$. Thus, on a tornadic day, the probability of a day with an EF3-EF5 tornado in that specific region equals to $\lambda_t$.

To find the best model for each region, we employ a genetic algorithm with 500 different initial models and we retain the same logistic regression model for each region (Table 5). Hereafter, we refer to these intensity prediction models as the IEOF models.

## Validation of LEOF and IEOF models

To validate the LEOF and IEOF models, we also consider competing models with more traditional tornado-related covariates such as CAPE, CP, SRH, CIN, 4LFTX [21, 28] as well as the

**Table 5. Summary of the fitted logistic regression models classifying strong/weak tornado days in each of the four regions.** For each region, the table has only the estimates of the significant predictors and their corresponding standard error (**Std. Error**).

| Coefficient (Std. Error) | R1 | R2 | R3 | R4 |
|---|---|---|---|---|
| Intercept | -3.389 (0.326) | -3.898 (.376) | -2.639 (0.249) | -2.783 (0.214) |
| $Temp - Rhum_3$ | -0.020 (.005) | | | |
| $Temp - Rhum_5(MSS)$ | 0.034 (.008) | | | |
| $Temp - Rhum_7(DLM)$ | 0.024 (.009) | 0.050 (.009) | 0.046 (.012) | |
| $Temp - Rhum_9$ | | | 0.047 (.012) | |
| $Temp - Rhum_{12}^2$ | -0.003(.001) | | | |
| $Temp - Rhum_{14}$ | | 0.039 (.014) | | |
| $Temp - Rhum_{15}^2$ | | | 0.003 (.001) | |
| $Wind_1$ (HSD) | 0.015 (.003) | 0.021 (.003) | | 0.015 (.002) |
| $Wind_2$ | 0.011 (.004) | | | |
| $Wind_2^2$ | | -0.012 ($<$0.001) | | |
| $Wind_6$ | | 0.019 (.005) | | |
| $Wind_7$ | | | 0.017 (.006) | |
| $Wind_{12}^2$ | | | | 0.002 (.0007) |
| Month April | | 0.734 (.028) | | |
| Month May | 0.962 (.245) | 1.391 (.343) | | |
| $Tropopause_2$ | | | 0.273 (.001) | |
| $Tropopause_2^2$ | | | -0.067 (.021) | |

GWO phases which have been previously shown to affect tornado occurrence [76]. Further, to validate the logistic regression, we consider a much more flexible random forest model [77] with EOF covariates. The random forest is a nonparametric regression technique that estimates probabilities and can account for the possible nonlinearity (if it exists) between the response and the different explanatory variables. Specifically, for each region we consider the following three competing models:

- Random Forest model: Random Forest model with the same 77 covariates considered for searching for the LEOF and IEOF models. We fit the random forest model using the R package randomForest [78].

- CP & SRH model: logistic regression model with CP and SRH as predictors. We also consider the inclusion in this model of seven indicator (dummy) variables representing the first seven phases of GWO. In addition, we also consider the first 11 months indicator variables (seasonal indicators) to account for seasonality, but their inclusion did not improve performance.

- Indices model: logistic regression model that uses as predictors CAPE, CIN, 4LFTX, CP, SRH, GWO indicators and all possible interactions.

For the CP & SRH and indices models, we use stepwise selection to select the best combination of the predictors with the highest BIC. To compare the competing models, we perform a Monte Carlo leave–20%–out cross validation with 1000 randomly drawn pairs of training and testing datasets. First, for the LEOF models, each training dataset consists of about 80% of the significant tornado days (cases) and their corresponding control days. In addition, each testing dataset contains the remaining tornado days and a number of non-tornado days that keeps the tornado incidence rate in the testing dataset close to actual rates in the original data. For

example, because the incidence rate of occurrence of a day with a significant tornado is 17%, we consider a testing dataset with incidence rate of 17%. Second, for the IEOF models, each training dataset consists of 80% of the weak and strong tornado days. The remaining 20% of the data is used for testing for each specific region.

### Study of temporal trends for the atmospheric EOF modes

For the atmospheric EOF modes that are statistically significant in either the LEOF or the IEOF models, we investigate whether any temporal trends exist. Specifically, we compute posterior probabilities for each of the long-term trend models (See Supplementary Materials for detailed procedures).

### Relative risk ($RR_j$) and the impact of each predictor on tornado probability through time

To measure the impact of the temporal change of each predictor (atmospheric EOF mode) on the tornado probability, we compute a conditional relative risk (RR) [63] that measures the change through time in the probability of strong tornado days. Here, we consider this measure of risk relative to temporal changes in the predictor with respect to a baseline period that we take to be the first five years of our analysis from 1980 to 1984. To measure the impact of the temporal change of a specific predictor $j$, the relative risk on day $t$ due to predictor $j$ is defined as

$$RR_{tj} = \frac{P_t}{P_{tj}^0},$$

(3)

where $P_t$ is the probability of a significant tornado day according to the LEOF models computed using the values of all predictors observed at 1800 UTC on day $t$. In contrast, $P_{tj}^0$ is the probability computed using for predictor $j$ its median value observed on the same month of the baseline period. For all the other predictors, $P_{tj}^0$ uses their observed values on day $t$.

## Results

### Framework of significant tornado cases and matched controls

To study the probability of strong tornado day based on the environment, we select meteorological variables at time steps before the first registered strong tornado for both the cases and the corresponding controls. Hence, on each tornado day the considered values of these meteorological variables correspond to time points from 0 to 3 hours before the first registered strong tornado. Table 1 shows that the southern regions (R1 and R2) have higher rates of occurrence of significant tornado days (2.42% and 2.4%) in comparison to the northern regions (R3 and R4) with rates of 1.82% and 1.83% respectively. In addition, the southern regions have registered 135 and 132 days with strong tornadoes (EF3-EF5) whereas the northern regions have registered 101 and 90 days respectively. Therefore, strong tornadoes have been more frequent in the southern regions (R1 and R2) than in the northern regions (R3 and R4).

### Selected EOF modes

We retain atmospheric EOF modes that explain about 83% and 69% of the total variability in temperature/relative humidity and wind data respectively (Tables 2 and 3). Specifically, we retain 15 TRH modes and 15 WC modes. Similar to EOF analysis of tornado proximity

soundings [79], the resulting EOF modes provide important information about the patterns of correlated variability (second-order behavior) of the meteorological variables on tornado and non-tornado days. In other words, each EOF mode is a data-based index that consists of a three-dimensional (latitude, longitude, and altitude) weighted average of the corresponding meteorological variables (TRH or WC).

Next, we discuss in more detail six of these EOF modes due to one or both of two reasons: 1) the mode appears in the LEOF and/or IEOF models through statistically significant main effects and 2) the mode exhibits a statistically significant temporal trend. Four of these important EOF modes are related to temperature and relative humidity, and two are related to wind (Fig 2).

**Stratospheric forcing mode.** The first intriguing mode (Fig 2a) mainly involves stratospheric temperature and relative humidity (200–10 hPa) with smaller contributions from the troposphere. A positive value of this EOF mode is associated with a warm and dry stratosphere. Therefore, we name this EOF mode the Stratospheric Forcing (STF) mode. STF is significant for predicting the occurrence of a significant tornado day according to the LEOF models over the western and southern regions (R1, R2 and R3) (Table 4). A moist troposphere has been linked with tornado activity in numerous previous studies (e.g. [19, 79, 80]). To our knowledge, a link between stratospheric temperatures and tornado days is not well established. However, there is a well-known relationship between stratospheric intrusions of high potential vorticity (PV) air and tropospheric dynamics. This relationship can be captured succinctly using a PV framework [81, 82]. Nielsen-Gammon and Gold (2008) used idealized numerical experiments to elucidate the role of upper tropospheric PV anomalies in creating a favorable environment for severe weather [39, 40]. Specifically, they found that upper-level PV anomalies produce decreased static stability beneath and adjacent to the PV anomaly and increased vertical wind shear surrounding the PV anomaly. Additionally, the PV perturbation led to changes in CAPE that would be favorable for severe weather. In the present study, the STF mode appears to be linked to the upper-tropospheric PV perturbation described in Nielsen-Gammon and Gold (2008) [40]. To illustrate this link, we construct composites of the dynamical tropopause (DT) pressure for the ten most positive (Fig 3a and 3b) and ten most negative (Fig 3c and 3d) analysis times from the entire 1980–2017 dataset for the southern regions (R1 and R2). Similar to previous studies [83–86], we define the DT as the surface at which potential vorticity is equal to 2 PV units (PVU). When the STF mode is positive, a strong PV anomaly extends southward from Canada into the Southern Plains region, as evidenced by the lowering of the dynamical tropopause across the central United States (Fig 3a and 3b). When the STF mode is negative, the DT pressure pattern is more zonal, and the Southern Plains are under the influence of ridging as indicated by a higher tropopause (Fig 3c and 3d). In a recent study of the environmental conditions of tornadoes outbreaks in Oklahoma, Illinois, and Alabama, Bray et al. (2021) found a significant association between tropopause polar vortices (TPV) and tornadic occurrence especially during the spring season [41]. Our results demonstrate a significant statistical linkage between stratospheric forcings and tornadic occurrence over a synoptic-scale spatial domain of the central United States.

**Moist static stability mode.** The next TRH mode is associated with moist static stability (MSS), since it measures contrast between relative humidity in the lower troposphere and relative humidity in the upper troposphere (Fig 2b). In conditions with moist lower tropospheric levels close to the surface and dry mid tropospheric altitudes would result in reduced moist static stability that favors formation of tornadoes. The MSS mode is statistically significant in the LEOF models for the first three regions (Table 2; R1, R2, R3). Additionally, the MSS mode has a significant decreasing trend over the western Midwest region (R3).

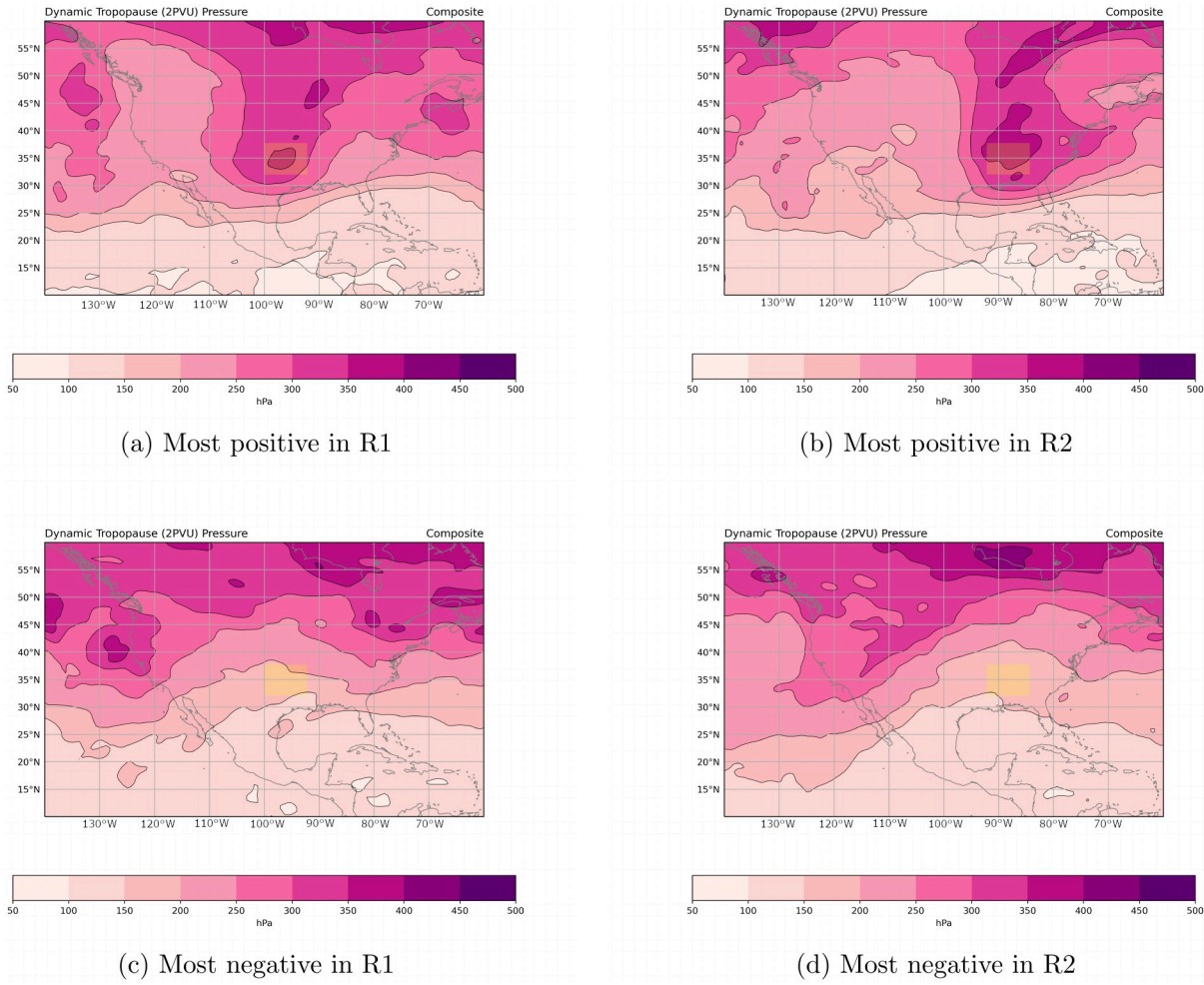

**Fig 3.** Composite maps of dynamical tropopause (2 PVU surface) pressure associated with the stratospheric forcing (STF) mode for times corresponding to (a) the ten most positive values in R1, (b) the ten most positive values in R2, (c) the ten most negative values in R1 and (d) the ten most negative values in R2. The yellow-shaded box is the corresponding study region.

**Dry static stability mode.** The third TRH mode measures mainly contrasts of atmospheric temperature with height (Fig 2c). Throughout the troposphere, this mode is associated with dry static stability. Larger values of this contrast correspond to more atmospheric instability. In addition, there is some east-west variation in moisture that may be related to an air mass transition across a frontal zone. To distinguish this EOF mode from the former MSS mode, we will refer to this EOF as the dry static stability (DSS) mode (Fig 2c). The DSS is statistically significant in the LEOF models for all four regions (Table 2). However, the DSS does not show significant temporal trends through time in any of the study regions (Table 6).

**Dryline mode.** The last atmospheric temperature/humidity mode corresponds to a more traditionally acknowledged atmospheric forcing associated with tornado-producing thunderstorms. We identify this temperature/relative humidity mode as the dryline mode (DLM), since it measures east-west contrasts of relative humidity in addition to atmospheric temperature contrasts. The DLM is significantly related to the occurrence of significant tornado days over the four study regions with positive coefficients in the LEOF models. According to its loadings (Fig 2c), the DLM increases with warm tropospheric temperature and relatively dry

**Table 6. Summary of the estimated coefficient for selected EOF modes in the LEOF models and the current temporal trend with its rate per decade for each region.** Only modes with significant temporal changes are included.

| EOF | STF | | MSS | | DLM | | ACM | |
|---|---|---|---|---|---|---|---|---|
| Region | Estimated coefficient | Current trend (rate/decade) | Estimated coefficient | Current trend (rate/decade) | Estimated coefficient | Current trend (rate/decade) | Estimated coefficient | Current trend (rate/decade) |
| R1 | 0.044 | -0.206 | 0.030 | Not signif | 0.106 | 0.095 | Not signif | -0.030 |
| R2 | 0.023 | 0.035 | 0.020 | Not signif | 0.091 | 0.107 | 0.026 | -0.039 |
| R3 | 0.022 | -0.124 | 0.049 | -0.055 | 0.064 | 0.071 | 0.024 | -0.030 |
| R4 | Not signif | 0.061 | Not signif | -0.057 | 0.065 | 0.072 | 0.027 | -0.035 |

air on the west side of the study region and moist air on the east side. These density differences in relative humidity (east-west) enhance the formation of a dryline with thunderstorms along and in advance of this line [87, 88].

**Horizontal wind speed and direction mode.** In terms of the wind EOF modes, one of the most important modes measures the horizontal wind speed and direction (HSD) (Fig 2d) over the study regions. According to the LEOF models, HSD affects the probability of the occurrence of significant tornado days in the four study regions. Specifically, we note that days with significant tornadoes have large positive values of HSD. When HSD is positive, there is a trough located downstream (to the east) of the domain, and when HSD is negative, there is a trough located upstream (to the west) of the domain. In addition to the trough location, HSD features positively tilted (southwest to northeast) troughs in both its positive and negative mode (not shown).

**Ageostrophic circulation mode.** The second important wind mode is an "ageostrophic circulation" mode, which we name ACM. This mode distinguishes between jet streak circulations that are more versus less favorable for large-scale rising motion. A jet streak is a wind speed maximum in the upper levels of the atmosphere, usually maximized around the 200–300 hPa level. When a simple four-quadrant model is applied to a straight jet streak, the left exit and right entrance quadrants are the most favorable locations for large-scale upward motion [89, 90]. In three of the four regions (R2, R3, R4), the ACM mode is a significant predictor in the LEOF models (Table 4). For each of these three regions, Fig 4 shows the upper-level jet stream configuration associated with positive and negative phases of the ACM mode, constructed by calculating the composite winds associated with the ten most positive and ten most negative observation times for this EOF mode. For regions R2, R3 and R4, we observe that the positive phase is associated with a jet stream configuration that is favorable for upward motion within the respective region (Fig 4a–4c). For example, there are two jet streaks observed for R2 (Fig 4a) with the R2 region (shaded) located in the left exit and right entrance regions of the upstream and downstream jet streaks, respectively. Interestingly, for R2 and R3, the ACM mode extracts a particular situation in which two jet streaks are located near the region of interest (Fig 4a and 4b). Such a coupled jet streak pattern is known to locally enhance vertical motion, and has been associated with increased precipitation rates, rapidly developing cyclones, and severe weather [91–94]. More recently, Kelnosky et al. (2018) [95] found that a coupled jet streak configuration was correlated with stronger tornado outbreaks. A coupled jet streak may also be more important in Southeastern U.S. (e.g., R2) tornado outbreaks compared with Southern Plains (e.g., R1) tornado outbreaks [95, 96]. For R4, there is a single jet streak, and the region of interest is positioned in the left exit region, which is also favorable for rising motion (Fig 4c). The negative phase of ACM is associated with weaker jet streaks and/or less favorable positioning for rising motion (e.g., right exit region for R2; Fig 4d).

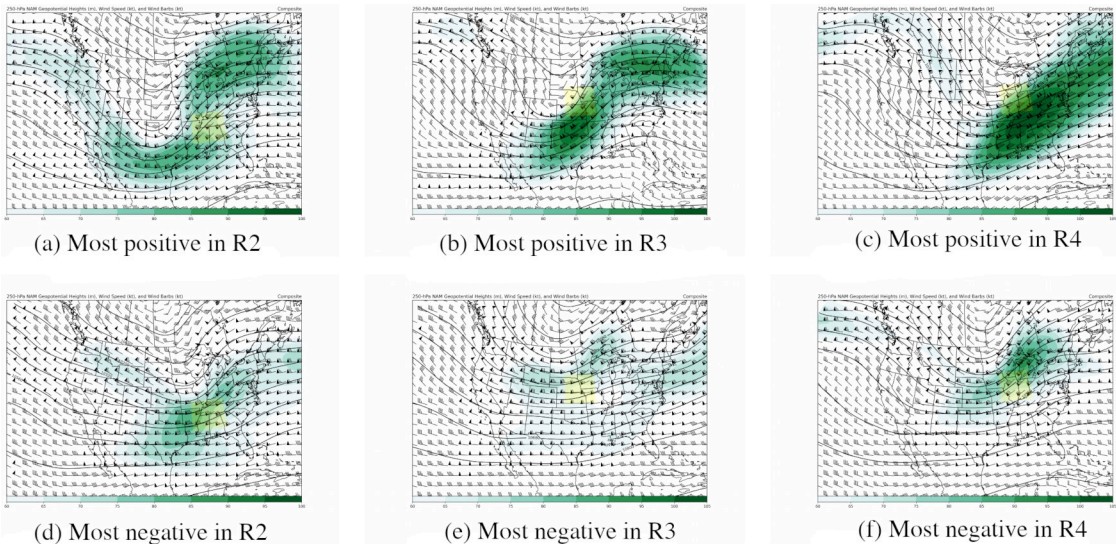

**Fig 4. Composite maps of 250 hPa geopotential heights (contoured), 250 hPa wind speed (shaded) and 250 hPa wind direction (wind barbs) associated with the ageostrophic circulation mode (ACM) over regions R2, R3 and R4.** Each map corresponds to (a-c) the ten most positive values (d-f) the ten most positive values for regions (left) R2, (middle) R3, and (right) R4. The yellow-shaded box is the corresponding study region.

## Probability of significant tornado days and validation of the LEOF models

To shed light on possible reasons for the observed decrease in the number of tornado days, we analyze how the atmospheric EOF modes affect the probability of a significant tornado day for each of the four study regions. Specifically, we build a logistic regression model for the probability of occurrence of a significant tornado day that utilizes the atmospheric EOF modes as explanatory variables for each of the four regions separately. That is, each LEOF model is a regional-scale model that determines which tornadic weather conditions are associated to tornado occurrence within a specific region.

Table 4 shows that several EOF modes such as DSS, DLM and HSD are significantly associated to occurrence of significant tornado days over the four study regions. For all significantly associated EOF modes, the estimated coefficients are consistent in their signs through the four regions. However, the estimated coefficients may vary in magnitude spatially. This indicates that an EOF mode has the same type of effect (positive or negative) on the probability of occurrence of a significant tornado day, but with different effect magnitudes across the four regions. For example, the MSS mode has estimated coefficients of 0.03, 0.02 and 0.049 over the R1, R2 and R3 regions respectively. This means that larger values of MSS are associated with higher probability of occurrence of a significant tornado day. Meanwhile, MSS has the highest influence in the R3 region compared to the other regions. Similarly, the STF mode is significant in regions R1, R2 and R3 with positive slopes of 0.044, 0.023 and 0.022 respectively. This indicates the importance of stratospheric temperatures in these regions, especially in region R1 where the corresponding estimated coefficient is about twice the estimates in regions R2 and R3.

Some other candidate predictors are only statistically significant in a few regions. For example, the monthly detrended Niño-3.4 index is only significant over the southern regions R1 and R2 (Fig 4). We note that the relationship between ENSO and tornado occurrence in the United States is well established [e.g., see [38, 72]]. In addition, the first tropopause EOF mode

**Table 7. Summary of the Monte Carlo leave-20%-out validation.** Models comparison validating the LEOF and the IEOF models compared to the other competing models (Random Forest, CP & SRH and Indices). Based on the receiver operating characteristic (ROC) from the Mote Carlo (MC) validation results, the table reports the area under the curve (AUC %) for the different models. The LEOF and the IEOF models have better AUC on average and over each region compared to the other models.

| Model | Significant/NON tornadic day classification | | | | Strong/Weak tornadic day classification | | | |
|---|---|---|---|---|---|---|---|---|
| Region | LEOF | Random Forest | CP & SRH | Indices | IEOF | Random Forest | CP & SRH | Indices |
| R1 | **92** | 89 | 78 | 89 | **76** | 67 | 52 | 65 |
| R2 | **93** | 91 | 89 | 92 | **80** | 70 | 72 | 76 |
| R3 | **94** | 85 | 68 | 91 | **73** | 63 | 64 | 66 |
| R4 | **93** | 88 | 82 | **93** | 72 | 66 | 71 | **73** |
| Average | **93** | 88 | 79 | 91 | **75** | 67 | 65 | 70 |

that consists of the average tropopause altitude pressure is significant only in region R1. Finally, the ACM mode is significant over regions R2, R3 and R4. In summary, each region has a regional specific LEOF model for the occurrence of a significant tornado day with some common modes over the four regions. However, some modes vary in their amount of influence spatially.

We validate our LEOF models against the competing models, as described in the methods section, by performing a Monte Carlo leave-20%-out cross validation (Table 7). Specifically, we compare the receiver operating characteristic (ROC) curves for the competing models from 1000 testing datasets. The ROC curves (not shown) of the LEOF models dominate the ROC curves of all competing models. Specifically, Table 7 shows that the average areas under the curve (AUC) for the different models are: CP & SRH model (AUC = 0.79), Random Forest model (AUC = 0.88), INDICES model (AUC = 0.91), and LEOF model (AUC = 0.93). Therefore, with the highest average AUC, our LEOF models provide more accurate estimates for the probability of significant tornado days in all four regions.

## Classifying the intensity of tornadic days and validation of the IEOF models

In addition to the LEOF models, we introduce another group of models to classify the intensity of tornadic days. In particular, for each region, we consider a logistic regression model to predict tornadic days as either weak (EF1-EF2) or strong (EF3-EF5). We do not include EF0 tornadoes because of a jump in their recorded numbers after the introduction of Doppler radar [97]. Including EF0 tornadoes would likely lead to the identification of spurious trends. We utilize the same candidate predictors used for the LEOF models, and similarly, we consider the data for the time step before the first tornado of the corresponding intensity for each region separately. Table 5 shows the significant predictors for each of the four regions. We notice that each region has its own combination of EOF modes associated with classification of tornadic intensity. None of the EOF modes are significant for all the four regions. However, DLM is significant in the southern and western regions (R1, R2 and R3).

To validate our IEOF models, we employ a Monte Carlo leave-20%-out cross validation with 1000 randomly drawn training and testing datasets. Table 7 shows the results for the AUCs for IEOF models compared to the competing models using traditional meteorological indices. The average AUC over the four regions for the different models which are CP & SRH model (AUC = 0.65), Random Forest model (AUC = 0.66), INDICES model (AUC = 0.70), and IEOF model (AUC = 0.75). Thus, our IEOF models provide more accurate results in predicting tornadic intensity over three of the four regions.

## Temporal trends of the statistically significant EOF weather modes

For the significant EOF weather modes in the LEOF and IEOF models, we study their temporal behavior. Table 6 provides, for each significant EOF weather mode, its estimated coefficient in the LEOF models as well as its current temporal trend given as a rate of change per decade for each region. MSS, DLM, and ACM exhibit statistically significant long-term temporal trends in the same direction, either increasing or decreasing, across the study regions. However, the corresponding rates vary across the four study regions (Table 6, Fig 5). For example, the DLM has been increasing through time with different rates over the four study regions (Table 6, Fig 5c–5f), with the highest rates over the southern regions (0.095 and 0.107 per decade in R1 and R2) compared to lower rates in the northern regions (0.071 and 0.072 per decade in R3 and R4). In addition, the ACM mode has been decreasing over regions R2, R3 and R4, where it has been significant in the corresponding LEOF models (Table 6, Fig 5g–5i). Our empirical finding of a decreasing trend of the ACM mode is consistent with the decrease in the mean wind shear predicted by global climate models [12, 23].

Finally, our analysis allows temporal trends that change through time. This flexibility captures an interesting long term trend for the STF mode. Specifically, while the STF mode decreases linearly through time in the western regions (Table 6, R1 and R3), in region R2 the STF mode decreases through time until the late 1990s but then starts increasing after around March 1999 (Fig 5a and 5b). This indicates that the possible impact of climate change on meteorological variables associated to tornado occurrence may vary spatially and temporally (Table 6).

## Impact of temporal changes in the EOF modes on the probability of a significant tornado day

The study of changes in the relative risk [63], as defined in the methods section, computed with the LEOF models allows us to estimate possible effects on tornado occurrence due to the identified temporal changes in the EOF modes in each region. Because the temporal changes in the EOF modes have different rates across the four regions, their impact on the probability of a significant tornado day varies across the regions. Here, we separate the possible effect of changes in each EOF mode on the probability of occurrence of a significant tornado day for each of the four regions.

Fig 6 shows the relative risk, with respect to the baseline years from 1980 to 1984, of a significant tornado day due to changes in relevant EOF modes through time. Of particular interest are changes in relative risk due to changes in the STF, DLM, and ACM modes.

First, temporal changes in the STF mode strongly impact the probability of significant tornado days in R1, R2, and R3 (Fig 6a–6c). For region R1, we notice a reduction by 55% on average in the period from 1997 to 1999 relative to baseline. Although there is an exception for the interval 1991–1993, the decaying temporal pattern of the relative risk is significant. Similar to region R1, for region R2 the reduction in the adjusted relative risk reaches its lowest at 34% in the period from 1997 to 1999. However, the relative risk of a significant tornado day has stabilized since 2000 due to the slight increase in the temporal trend of STF in region R2. Finally, for region R3 the adjusted relative risk has been decreasing and its lowest level was 29% on average in the period from 2000 to 2002. Therefore, the impact of changes in the STF mode has been at its maximum over region R1, and the impact is a reduced probability of tornado occurrence in the western Midwest, Southern Plains and southeastern U.S. regions.

Second, the DLM has been increasing significantly across the four regions (Fig 5c–5f). However, the rates of increase vary from region to region. The highest increase in the RR has

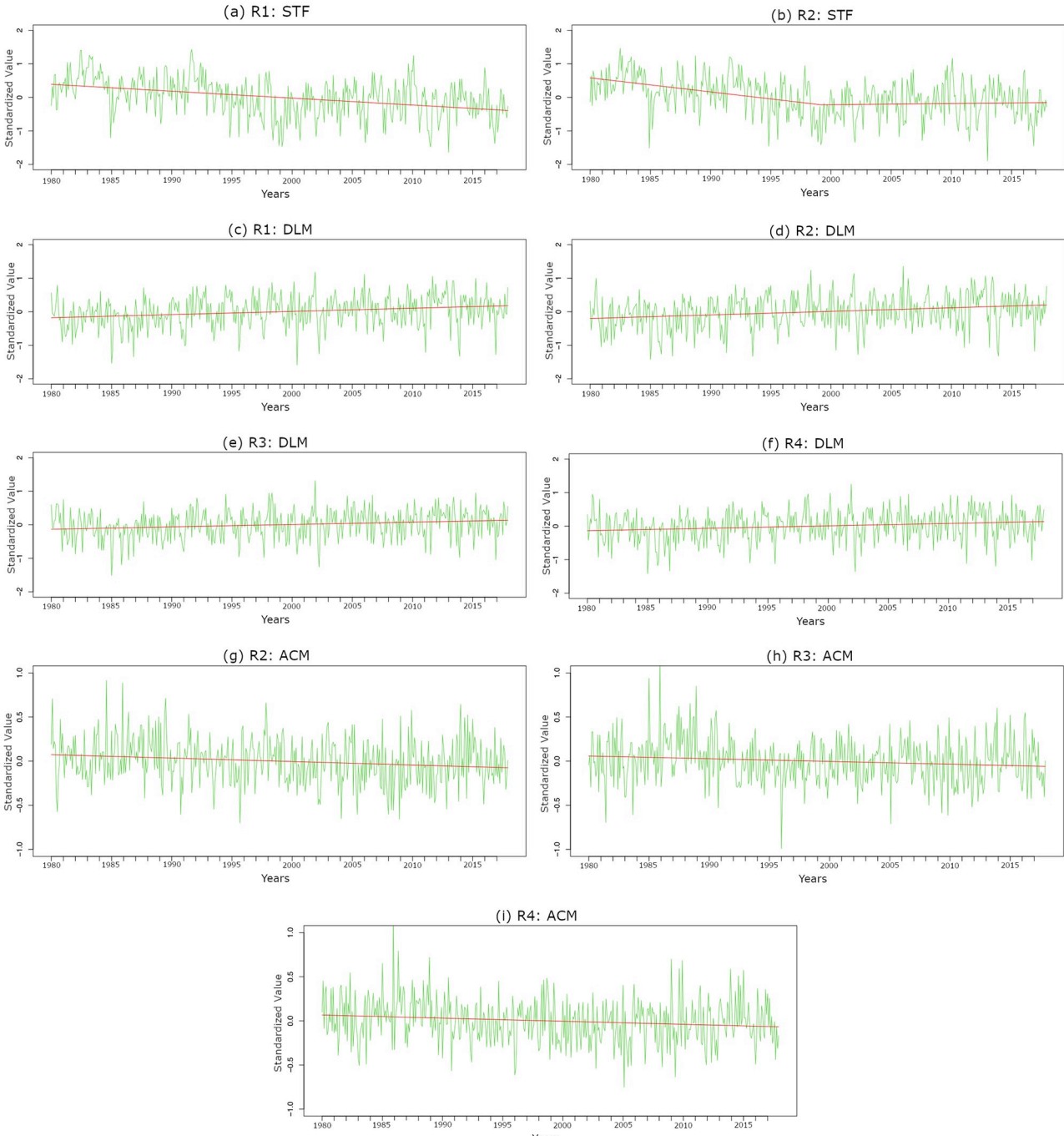

**Fig 5. Monthly deseasonalized series for STF, DLM, and ACM from 1980 to 2017 with the red lines indicating the estimated trends for specific regions.** Table 6 shows the estimated trends for the selected EOF modes for the four regions. (a) R1: STF, (b) R2: STF, (c) R1: DLM, (d) R2: DLM, (e) R3: DLM, (f) R4: DLM, (g) R2: ACM, (h) R3: ACM, and (i) R4: ACM.

been observed in R1 then R2 with rates on average 54% and 46% respectively in the period from 2012 to 2014. For R3, the relative risk has been increasing with a lower rate compared to the southern regions (R1 and R2). That increase in the relative risk reaches a maximum of 15% in the period from 2003 to 2005. Finally, region R4 has the lowest rate of increase in the relative

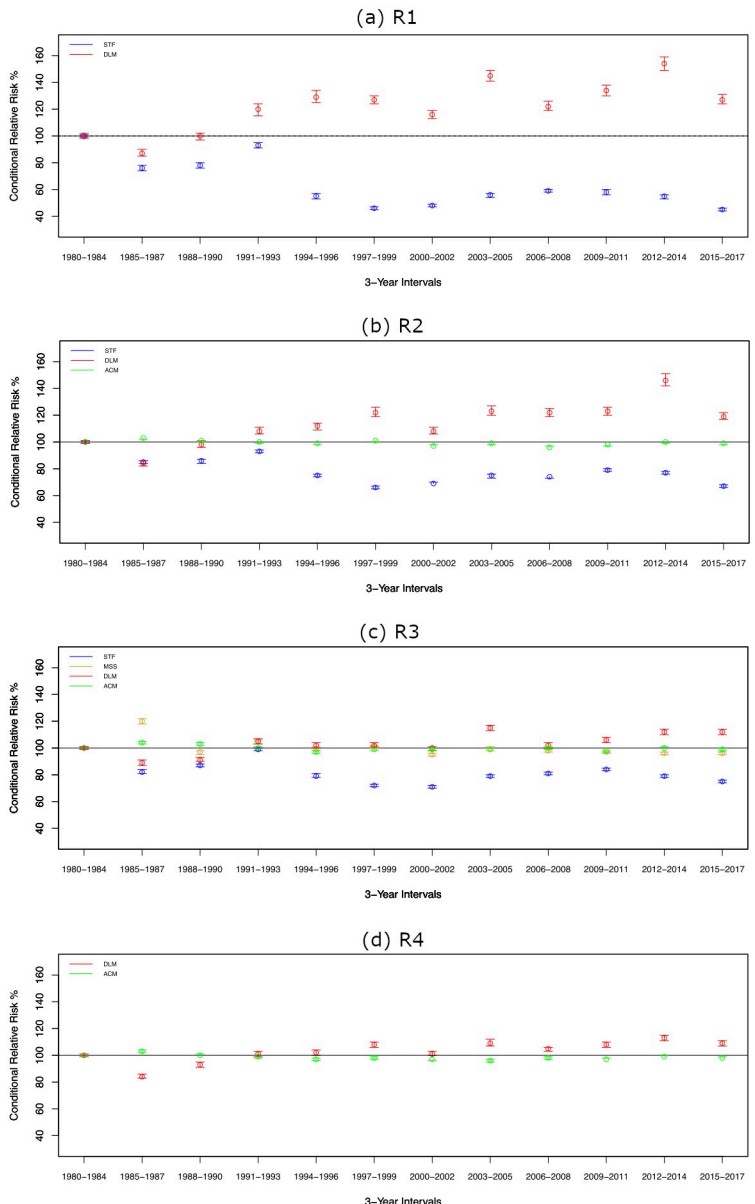

**Fig 6. Change in relative risk (RR) of a significant tornado day through time due to temporal changes in the EOF modes, including 95% confidence intervals for median RR of significant tornado day for each 3-years time period compared to baseline period (1980–1984) for selected modes in the LEOF models.** (a) R1, (b) R2, (c) R3, and (d) R4.

risk compared to the other regions with an increase of 13% on average in the period from 2012 to 2014.

Third, the ACM shows significant temporal trends over R2, R3 and R4. However, the corresponding impacts on the relative risk due to these changes are not as prominent as the influences due to changes in the STF or the DLM (Fig 6b–6d). Lastly, the MSS mode results in only meager changes in the relative tornado risk over time (Fig 6c). Other temperature/relative humidity modes contribute to more prominent changes in relative risk.

In summary, when compared to changes in the ACM and MSS, the long term trends observed in the DLM and STF modes have had a much stronger effect on the risk of significant

tornado days (Fig 6). Further, the effect of the decrease in the STF mode offsets the increase in the DLM in the southern regions (R1 and R2) (Fig 6a and 6b). In region R3, the effect of the decreasing trend of the STF mode swamps the modest increase in the DLM (Fig 6c). Finally, in region R4, these results suggest a slight increase in tornado risk mainly due to changes in DLM (increased frequency of dry line environmental forcings).

## Conclusion

We performed a statistical climatological study to identify long-term trends in temperature, relative humidity, and winds related to the occurrence of days with strong tornadoes in the Central, Midwestern and Southeastern United States. Specifically, we computed temperature-relative-humidity EOFs and wind EOFs using the MERRA-2 dataset. With these EOF modes, we built statistical models to predict significant tornado days (LEOF models) and to classify tornadic days as either weak or strong (IEOF models).

Our study identified five synoptic-scale ingredients that have previously been associated with severe weather environments: 1) a moist static stability (MSS) mode, 2) a (dry static stability (DSS) mode), 3) a dry line mode (DLM) associated with east-west relative humidity contrasts, 4) a horizontal speed and direction (HSD) mode that is related to the large-scale trough and ridge pattern, and 5) an ageostrophic circulation mode (ACM) that is related to the jet stream configuration. Of these five modes, three show statistically significant long-term trends: the MSS, DLM, and ACM. The MSS mode measures contrast between relative humidity in the lower troposphere and relative humidity aloft. It is statistically significant in the LEOF models in southern Plains (R1), the southeastern (R2) and the western Midwest (R3) regions, and shows a significant decreasing trend over the western Midwest region (R3). The DLM is statistically significant in the LEOF models for all four regions spanning the central, midwestern and southeastern U.S. In addition, our analysis shows that the DLM has statistically significant increasing temporal trends in all four study regions. The ageostrophic circulation mode (ACM) distinguishes between jet streak circulations that are more versus less favorable for large-scale rising motion. This mode is a significant predictor in the LEOF models in southeastern region (R2) and both midwestern regions (R3 and R4). In addition, the ACM mode has statistically significant decreasing temporal trends in regions R2, R3 and R4, which is consistent with previous studies based on global climate models that have predicted a decrease in the mean wind shear due to climate change.

A novel finding in our study is the importance of another synoptic-scale ingredient, a stratospheric forcing mode (STF), that influences the occurrence of strong tornado days. The STF is related to stratospheric temperature and is significant in the LEOF models for the Southern Plains region (R1), the southeastern region (R2) and the western Midwest region (R3). Further, we constructed composites of the dynamical tropopause (DT) pressure that demonstrate a link between the STF mode and the upper-tropospheric PV perturbation. These results indicate that the STF mode has important long-term trends: while the STF mode has been decreasing through time in the western regions R1 and R3, in region R2 the STF mode was decreasing through time until the late 1990s but then started increasing after 1999. This indicates that the possible impact of climate change on meteorological variables associated to tornado occurrence may vary spatially and temporally. An increase in the STF mode in R2 would support an increase in tornado frequency, which has been observed in this region [4–7].

There are a couple of important predictors in the LEOF models that do not show significant temporal trends. Specifically, the HSD mode, which measures whether a trough located downstream (to the east) or upstream (to the west) of the study region, is a very important predictor

in the LEOF models for all four study regions. However, HSD does not have statistically significant temporal trends in any of the study regions. In addition, the dry static stability mode (DSS) is also an important predictor in the LEOF models for all four study regions, but the DSS does not have statistically significant temporal trends in any of the study regions.

To contrast the impacts of the different EOF modes on the risk of significant tornado days, we performed an analysis of conditional relative risk. In general, the MSS and ACM modes are not leading to large changes in relative risk of tornado occurrence since 1980. In contrast, changes in STF and DLM have a much stronger impact on the risk of significant tornado days. Specifically, the dry line mode (DLM) is leading to higher relative risk in all four regions, but most strongly in the southern Plains and southeastern regions (R1 and R2). Meanwhile, changes in the stratsopheric forcing mode (STF) are partially or completely offsetting the increased tornado risk associated with increases in DLM in the Southern Plains (R1), the southeastern (R2) and the western Midwest (R3) regions. In the eastern midwest region (R4), which spans Illinois, Indiana, eastern Iowa, and northeastern Missouri, there is a slight increase in relative tornado risk that is mainly associated with the DLM. Collectively, this analysis shows that, when compared to changes in wind (e.g., ACM), changes in temperature and relative humidity (e.g., STF and DLM) have a much stronger impact on the risk of significant tornado days. Our results are consistent with other studies that have suggested that the climate change effects would lead to increased frequencies in favorable thermodynamic environments that exceed the decrease in the mean wind shear [20, 23, 97]. However, we recommend caution in interpreting these results due to 1) uncertainty in the underlying MERRA-2 reanalysis thermodynamic fields and 2) uncertainty as to whether this dominance of the temperature and relative humidity modes will continue into the future. Additional studies should be conducted to confirm that similar synoptic-scale environmental trends exist in other datasets (such as ERA5) and whether these trends lead to similar changes in relative tornado risk.

Finally, this research identifies several synoptic- to mesoscale ingredients that are favorable for tornado formation, including a stratospheric forcing (STF) mode and an ageostrophic circulation mode (ACM) that have been less studied. Although there is a well-known linkage between stratospheric forcings and upper troposphere dynamics [40], only a few studies have linked these dynamic forcings with severe weather [39, 40] and with tornado formation [41, 95]. Interestingly, Bray et al. (2021) used ERA-I data to identify tropopause polar vortices, whereas our study uses EOF analysis of MERRA-2 data to identify synoptic-scale ingredients which included the STF mode. These disparate data and methods yet similar results lend more confidence to the statistical results presented in this study. Future research should perform more in depth analyses into how stratospheric forcings, tropopause polar vortices, and jet stream circulations provide favorable environments for tornado outbreaks. Additionally, more research needs to identify whether these synoptic-scale ingredients are becoming more or less common at the regional scale as the climate warms.

## Supporting information

**S1 File.**
(ZIP)

## Acknowledgments

All the authors declare no competing interests. United States tornado report data are from NOAA's Storm Prediction Center http://www.spc.noaa.gov/wcm/. MERRA-2 data provided

by NASA Global Modeling and Assimilation Office (GMAO) http://disc.sci.gsfc.nasa.gov/mdisc/. GWO data, and NARR data provided by the NOAA/OAR/ESRL PSD, Boulder, Colorado, USA, http://www.esrl.noaa.gov/psd/.

## Author Contributions

**Conceptualization:** Mohamed Elkhouly, Marco A. R. Ferreira.

**Data curation:** Mohamed Elkhouly.

**Formal analysis:** Mohamed Elkhouly, Stephanie E. Zick, Marco A. R. Ferreira.

**Funding acquisition:** Marco A. R. Ferreira.

**Investigation:** Mohamed Elkhouly, Stephanie E. Zick, Marco A. R. Ferreira.

**Methodology:** Mohamed Elkhouly, Stephanie E. Zick, Marco A. R. Ferreira.

**Project administration:** Mohamed Elkhouly, Marco A. R. Ferreira.

**Resources:** Mohamed Elkhouly, Marco A. R. Ferreira.

**Software:** Mohamed Elkhouly, Stephanie E. Zick, Marco A. R. Ferreira.

**Supervision:** Stephanie E. Zick, Marco A. R. Ferreira.

**Validation:** Mohamed Elkhouly, Marco A. R. Ferreira.

**Visualization:** Mohamed Elkhouly, Stephanie E. Zick.

**Writing – original draft:** Mohamed Elkhouly.

**Writing – review & editing:** Mohamed Elkhouly, Stephanie E. Zick, Marco A. R. Ferreira.

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
