## [Decision Letter · Decision Letter 0]

22 Nov 2022

PONE-D-22-28676Long term temporal trends in synoptic-scale weather conditions favoring significant tornado occurrence over the central United StatesPLOS ONE

Dear Dr. Zick,

Thank you for submitting your manuscript to PLOS ONE. After careful consideration, we feel that it has merit but does not fully meet PLOS ONE’s publication criteria as it currently stands. Therefore, we invite you to submit a revised version of the manuscript that  fully addresses all the points raised during the review process by both reviewers.

We look forward to receiving your revised manuscript.

Kind regards,

João Miguel Dias, Ph.D.

Academic Editor

PLOS ONE

Journal Requirements:

a) You may seek permission from the original copyright holder of Figure 1 to publish the content specifically under the CC BY 4.0 license.  

Reviewers' comments:

Reviewer's Responses to Questions

**Comments to the Author**

1. Is the manuscript technically sound, and do the data support the conclusions?

Reviewer #1: Yes

Reviewer #2: Yes

2. Has the statistical analysis been performed appropriately and rigorously? 

Reviewer #1: I Don't Know

Reviewer #2: Yes

3. Have the authors made all data underlying the findings in their manuscript fully available?

Reviewer #1: Yes

Reviewer #2: Yes

4. Is the manuscript presented in an intelligible fashion and written in standard English?

Reviewer #1: Yes

Reviewer #2: Yes

5. Review Comments to the Author

Reviewer #1: General Comments

The manuscript “Long term temporal trends in synoptic-scale weather conditions favoring significant tornado occurrence over the central United States” (PONE-D-22-28676), submitted to PLOS ONE, deals with a relevant topic, it is generally well written and overall, it is interesting for a wide audience. However, some aspects are not clear and a number of minor, mostly formal corrections including typos and figure improvements, are needed. I would like to see it published but I do not think it should be accepted in its current form. Therefore I recommend a review following as guidance the items listed below.

Specific Comments

1. Page 2, last paragraph. When listing the variables considered (air temperature, relative humidity, etc.) it is not clear if specific levels or all levels of the MERRA-2 reanalysis data are considered.

2. Page 3, line 69. The text mentions ‘section 2’ here, but sections are not numbered. Perhaps, ‘the second section’ or something similar would be more appropriate.

3. Page 8, Table 4. For a given variable (column), please check if the same number of decimal digits are needed.

4. Page 11, line 332 (and 339 and other references similarly). Reference ‘Nielsen-Gammon and Gold’ is not numbered (as other references given in the text) – please check journal style citation format.

5. Page 11, line 341. Please correct typo: Figure ?? c-d.

6. Page 12, line 362. ‘Atmospheric Instability Mode’: if there is a ‘Moist Static Stability Mode’ should include the term ‘Dry’ or something similar, to avoid possible misunderstandings?

7. Page 12, Table 6. Despite 6 EOF modes are described in Results section b) only 4 are listed in this table. Perhaps I missed something but I think this should be clarified (maybe instead of using the term ‘noteworthy EOF modes’ in the table, you could simply call them ‘Selected EOF modes’ or something similar; alternatively, a larger table listing all 6 modes could be considered).

8. Page 13, line 409. Similarly, as item 4, reference unnumbered.

9. Page 14, lines 464-465. The no inclusion of EF0 cases affects previous results? If so, I think this should be explained earlier, when describing the datasets used. Otherwise, please clarify in text.

10. Page 15, line 498. Typo: please add blank space 6,R1 -> 6, R1

11. Page 16, Conclusion sections indicates that ‘5 synoptic-scale ingredients’ were identified – from the 6 EOF modes found (but only 4 selected to be shown in Table 6). I found this a bit confusing; perhaps authors could try to reformulate this explanation linking it to the previous discussion.

12. Page 17, last paragraph. Please make sure that all locations mentioned in the text (Illinois, Indiana, etc.) appear labeled in Figure 1 so readers not familiar with the area of study can follow the explanation.

13. Page 21. Please check Ref. 16 and 24 are apparently duplicated.

14. Page 23. Ref. 45: please rewrite the article title (now it is in upper case).

15. Page 23. Ref. 50, typo: andtornado -> and tornado

16. Page 24. Ref. 68, typo: unbalaced -> unbalanced ?

17. Page 25. Ref. 78: please rewrite authors names (now they are in upper case).

18. Please check figures resolution and quality (labels of fig. 2, 3, 5 and 6, etc.).

Reviewer #2: Overview: The authors present a well-written and timely manuscript on an important topic. I find no major issues with the data or analyses, and have only minor suggestions (mainly for the intro and conclusion).

Introduction: Incomplete representation of recent relevant literature (details below).

Line 7: Authors use reference 4 to support their claim of negative and positive trends in plains and eastern US. This study is built on numerous studies that came before it. Citing numerous studies from the body of literature from 2014-present would be stronger and would account for the numerous methods that have been used to illustrate these opposing trends.

Line 8-10: Authors state that the increase in the eastern US is mainly explained by increases in population density. I disagree with this statement. This reference shows that population density combined with radar explain less than 30% of the variability.

Line 9: Should this say "while decreases in the US Plains..."?

Line 28-30: The literature does illustrate regional variability in the trends of proxies, but there is agreement across studies in the regional variability (important point that I think is missing in this characterization).

Line 341: Should this reference be to Figure 3?

Line 343: Please cite the studies that this result is similar to.

Line 559-563: Is the STF mode missing in this list?

Line 587-589: It would be insightful to discuss these trends in the context of documented regional tornado trends.

6. PLOS authors have the option to publish the peer review history of their article (what does this mean?). If published, this will include your full peer review and any attached files.

Reviewer #1: No

Reviewer #2: **Yes: **Todd W Moore

---

## [Author Response · Author response to Decision Letter 0]

6 Jan 2023

Please see attached Response to Reviewers file.

---

## [Decision Letter · Decision Letter 1]

20 Jan 2023

Long term temporal trends in synoptic-scale weather conditions favoring significant tornado occurrence over the central United States

PONE-D-22-28676R1

Dear Dr. Zick,

We’re pleased to inform you that your manuscript has been judged scientifically suitable for publication and will be formally accepted for publication once it meets all outstanding technical requirements.

Kind regards,

João Miguel Dias, Ph.D.

Academic Editor

PLOS ONE

Additional Editor Comments (optional):

Reviewers' comments:

Reviewer's Responses to Questions

**Comments to the Author**

1. If the authors have adequately addressed your comments raised in a previous round of review and you feel that this manuscript is now acceptable for publication, you may indicate that here to bypass the “Comments to the Author” section, enter your conflict of interest statement in the “Confidential to Editor” section, and submit your "Accept" recommendation.

Reviewer #1: All comments have been addressed

Reviewer #2: All comments have been addressed

2. Is the manuscript technically sound, and do the data support the conclusions?

Reviewer #1: (No Response)

Reviewer #2: (No Response)

3. Has the statistical analysis been performed appropriately and rigorously? 

Reviewer #1: (No Response)

Reviewer #2: (No Response)

4. Have the authors made all data underlying the findings in their manuscript fully available?

Reviewer #1: (No Response)

Reviewer #2: (No Response)

5. Is the manuscript presented in an intelligible fashion and written in standard English?

Reviewer #1: (No Response)

Reviewer #2: (No Response)

6. Review Comments to the Author

Reviewer #1: (No Response)

Reviewer #2: Thank you for addressing my comments and those of the other reviewer and editor. I look forward to seeing this paper in the literature.

7. PLOS authors have the option to publish the peer review history of their article (what does this mean?). If published, this will include your full peer review and any attached files.

Reviewer #1: No

Reviewer #2: **Yes: **Todd W Moore
